# [RE] Counterfactual Generative Networks

## Reproducibility Summary

**Scope of Reproducibility**

In this paper, we attempt to verify the claims that the paper [11] makes about their proposed CGN framework that decomposes the image generation process into independent causal mechanisms. Further, the author claims that these counterfactual images improves the out-of-distribution robustness of the classifier. We use the code provided by the authors to replicate several experiments in the original paper and draw conclusions based on these results.

**Methodology**

We use the same hyperparameters and architecture as mentioned in CGN [11]. We use the PyTorch code from the authors' publicly available repository. We make several changes to their code for the MNIST datasets since it gives spurious results/errors. Since we use ImageNet 1000 as a replacement for the ImageNet dataset, we modify the code accordingly. We reproduce tables 1-6 from CGN [11] paper, excluding results for models from other papers.

**Results**

We validated each of the author's claim through the experiments given in the original paper and few additional experiments of our own. Overall, we found many experiments yielding identical results while some deviations were observed with both the Counterfactual Generative Network and the subsequent classification task. We were able to explain most of these deviations through our additional experiments while some couldn't be validated due to computational limitations.

**What was easy**

Overall, clear environment setup instructions, well working code and availability of pretrained CGN models for both datasets proved valuable to validate the authors' claim.

**What was difficult**

Some experimental details were not reported in the original paper which made validations time consuming. ImageNet based experiments were replaced with ImageNet-1k(mini) due to the computational limitation which made it difficult to validate the author's original claims. Pre-trained classification models could have proven helpful in this case, but were unavailable, which meant we had to train the classifier from scratch. Code changes were required to obtain baseline results which was tedious considering different code architecture was implemented for MNIST & ImageNet.

**Communication with original authors**

We emailed the authors regarding inception score, MNIST dataset hyperparameters and ImageNet hyperparameters. We are awaiting a response from their end.

Code available at `https://anonymous.4open.science/r/re_counterfactual_generative-E18F`

# 1   Introduction

Neural Networks (NNs) have become ubiquitous in machine learning due to their predictive power. However, a shortcoming of NNs is their tendency to learn simple correlations that lead to good performance on test data rather than more complex correlations that generalise better. This shortcoming is apparent in the task of image classification, where NNs tend to overfit to factors like background or texture. To address this shortcoming, [11] proposes a method of generating counterfactual images that prevent classifiers from learning spurious relationships.

The authors take a causal approach to image generation by splitting the generation task into independent causal mechanisms. The authors considered three separately learned Independent Mechanisms (IMs) to generate shapes, textures and backgrounds for an image. For the MNIST setting, all IM specific losses are optimized end-to-end from scratch, while in the ImageNet setting, each IM is initialized with weights from pre-trained BigGAN-deep-256[1]. The counterfactual image is then generated by passing the result of each IM to a deterministic composer function.

In this report, we use the publicly available code provided by the authors to reproduce the results of the paper and validate the authors' claims. In this endeavour, we made modifications to the code to determine the efficacy of their generative model and validate its impact on improving the out of distribution robustness of a classifier.

# 2   Scope of reproducibility

In this report, we investigate the following claims from the original paper:

1. Generating high-quality counterfactual images that decompose into independent causal inductive biases, these mechanisms disentangle object shape, object texture and background

2. Using counterfactual images improves the shape vs texture bias which is an inherent problem of deep classifiers

3. Using counterfactual images improve the out-of-distribution robustness for the classifier during the classification task

4. The Generative model can be trained efficiently on a single GPU with the help of powerful pre-trained models

We attempt to reproduce the experiments from the paper [11] and perform exploratory analysis on the above mentioned claims. We propose using an extra loss function to mitigate some of the shortcomings during counterfactual generation process and generate heatmap plots to study the classifier behaviour.

# 3   Methodology

Alex *et al.* [11] propose a Counterfactual Generative Network (CGN) framework to generate high-quality counterfactual images, which can be used to train invariant classifiers. The architecture of a CGN is composed of three IMs that are trained to generate backgrounds, shapes, and textures. Each IM is provided with a label. The task of the invariant classifier is to predict the label of a specific IM, regardless of the labels of the others. In conjunction with the composer function, the use of counterfactual images generated by the three IMs prevents the classifier from learning spurious relationships that arise from training on a natural dataset only.

The architecture of the CGN consists of a GAN as the backbone of each IM. Each IM samples random noise $\mu \sim N(0,1)$, along with an independently sampled label to generate samples. The output $x_{gen}$ is generated using an analytical function from the Composer 'C',

$$x_{gen} = C(m, f, b) = m \otimes f + (1 - m) \otimes b$$

where 'm' is the mask (alpha map), f is foreground and b is background. $\otimes$ denotes the element wise multiplication.

The losses $\mathcal{L}_{rec}$ $(x_{gt}, x_{gen})$, $\mathcal{L}_1$ reconstruction loss, $\mathcal{L}_{perceptual}$ as shown in Fig. 1 are used to improve the quality of generated images. Once the CGN is trained, u and y are randomized per mechanism such that new counterfactual $x_{gen}$ are generated. Furthermore, hyperparameters such as CF ratio (the ratio indicates how many counterfactuals are generated per sampled noise) can be used to control the number of samples that are being generated. These samples are then used to train the classifier and evaluated on the corresponding test set.

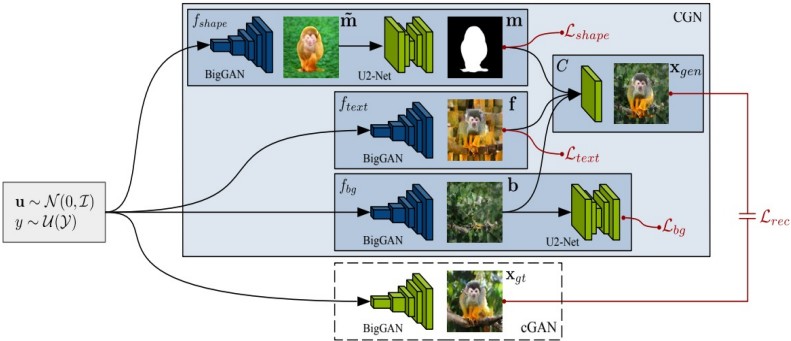

Figure 1: Architecture diagram from [11] for ImageNet [4] dataset.We observe that the architecture consists of $f_{bg}$, $f_{texture}$, $f_{shape}$ to assist with the generation of $x_{gen}$. A powerful pre-trained Biggan-256 [1] is used to images from noise for each of the independent mechanisms. The shape and background are extracted with the help of a pre-trained U2-net [8], while texture is obtained by minimizing perceptual loss between the foreground ($f_{text}$ and a patch grid obtained from the value within the mask). The composer is analytically defined which uses alpha blending to generate the counterfactual $x_{gen}$. Components with trainable parameters are 'green' and without are 'blue'.

## 3.1 Model descriptions

The ImageNet variant follows the architecture that is illustrated in Fig. 1. The MNIST variant applies a simpler architecture by applying a second texture mechanism rather than a background mechanism.

## 3.2 Experimental setup and code

We use the datasets mentioned in [11], excluding ImageNet [3] due to limited resources and computational constraints.

| Dataset | Description |
|---|---|
| Colored MNIST | Consists of digits in red or green. |
| Double Colored MNIST | Consists of more varied backgrounds and digits than Colored MNIST. |
| Wildlife MNIST | An attempt to build MNIST [6] closer to the ImageNet[3], texture was added as a bias to the data. The ten digits of the striped texture class encode the foreground lables and the background is labelled with the with the texture class 'veiny'. |
| ImageNet-1k(mini) | Subset of the ImageNet-1k[10],available here[1] that contains 34745 images in train set and 3923 for validation set, each split among 1000 classes individually. |

Table 1: Datasets used

For all the experiments, we make use of standard dataset splits akin to the CGN paper [11]. Considering the computational constraint to train a classifier on ImageNet[4], we used the pre-trained CGN to generate counterfactual images and trained a classifier on ImageNet-1k(mini) and mini-imagenet datasets.

## 3.3 Hyperparameter search

We found that the hyperparameters provided by the authors were stable, and so we did not conduct a hyperparameter search in this report.

## 3.4 Computational requirements

All models are run on Nvdia GTX1080Ti GPUs (11Gb VRAM). For the MNIST datasets, training a CGN and a classifier each took approximately one hour.

---

[01]https://kaggle.com/ifigotin/imagenetmini-1000

# 4 Results

A lack of compute power prevented us from replicating the experiments on ImageNet. As a workaround, we limit ourselves to verifying the results using the ImageNet-1k(mini) dataset. This is beneficial because it extends the results of the paper and evaluates the method on a new dataset, and ensures that results can be reproduced with limited resources by referring to our report/code and the CGN paper.

## 4.1 Results reproducing original paper

### 4.1.1 Can Image generation process be decomposed into independent causal inductive biases effectively?

We begin the experiment by training a CGN on the three variants of the MNIST dataset. We observe in Fig. 2 that the digits in case of colored MNIST dataset lose their shape when reconstructed, whereas for double colored and wildlife MNIST, the digits look much better. Since we do not clearly understand why the shape in Colored MNIST is poor, we generated a mask timeline to verify any patterns. Fig. 3a details the same. Further analysis on this was conducted and recorded in 4.2. We also propose an additional loss function to help mitigate this problem.

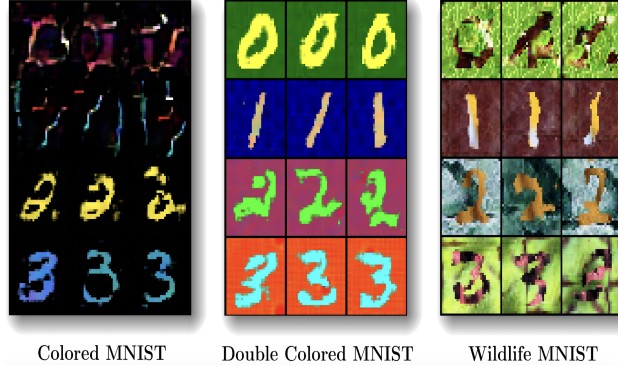

Colored MNIST      Double Colored MNIST      Wildlife MNIST

Figure 2: For brevity, we display only first 3 digits that were generated by training from scratch by us for the given three MNIST datasets.

**Quality of Counterfactual Images on ImageNet-1k**

To quantify the quality of the composite images produced by the CGN, the authors calculate the inception score (IS). The details of the IS calculations (inception model used, number of images used) were not mentioned in the paper. In an attempt to recreate the results regarding IS, we use the OpenAI implementation [1]. We plot the results of IS vs the number images using 10 splits in Fig. 9. We observe the IS converges to an IS of 198.

We made use of the pre-trained CGN trained on ImageNet-1k that was present as part of the codebase to generate counterfactual images. Since there is no quantitative way to measure the quality of counterfactual images, we reproduced the images given in the original paper. We achieved a similar quality of counterfactual images but also noted deviations. Fig. 7 shows all the images that were given in the original paper. A deviation in the mask is observed for the class 'Agaric' and 'Cauliflower'. The difference in the images to the original paper prompted us to collect the classes with poorer counterfactual images to observe any patterns.

Fig. 8 is generated from the pre-trained CGN that have a low quality of images picked from random classes. Since the analysis is qualitative, we relied on the realism of the counterfactual compared to original images from that class. Images under the classes 'Cliff dwelling' 'American Chameleon' suffer from Texture-background entanglement resulting in the counterfactual with no subject. On the other hand, the images under the class 'Goldfinch', 'Junco' suffer from reduced realism due to linear constraints applied on the composer.

### 4.1.2 Impact of counterfactual images towards shape-bias of the classifier

**Experiments conducted with ImageNet-1k(mini) dataset**

---

[1]https://github.com/nnUyi/Inception-Score

In order to identify the impact of shape bias on the classifier, we made use of the proposed architecture for the classifier ensemble that included 3 different heads. The ensemble includes a pre-trained classifier(we made use of Resnet-50) as the backbone, while attaching 3 different heads to it. Each head controls the variance with respect to one of the 3 independent mechanism(Shape, Texture, Background) which are individually trained from scratch. The result from these heads are averaged to get the prediction accuracy of the classifier ensemble.

The results in Table 2(a) for ImageNet-1k(Mini) showed a considerable deviation. The shape bias is marginally lower compared to the baseline result while the texture bias is high. The reduction in the shape bias could be due to the smaller dataset that we are using. Since this is ambiguous to validate the original claim we conducted additional experiments which are detailed in section 4.2.

| Dataset | Shape Bias |
|---|---|
| ImageNet-1k | 48.1% |
| ImageNet-1k + CGN/Shape | 47.00% |
| ImageNet-1k + CGN/Text | 37.01% |
| ImageNet-1k + CGN/Bg | 47.02% |

(a) Impact on shape bias

| Dataset | IN-9 | Mixed Same | Mixed Rand | BG-Gap |
|---|---|---|---|---|
| ImageNet-1k | 17.27% | 6.37% | 7.65% | 1.28% |
| ImageNet-1k + CGN | 18.2% | 14.05% | 12.35% | 1.7% |

(b) Out-of-distribution accuracy for ImageNet variants

| Dataset | Top-1 Train Accuracy | Top-5 Train Accuracy | Test Accuracy |
|---|---|---|---|
| ImageNet-1k(mini) | 91.27% | 97.35% | 73.12% |
| ImageNet-1k(mini) + CGN | 90.32% | 97.24% | 11.36% |

(c) Train and Test accuracies for ImageNet-1k(mini) with Resnet-50 backbone

Table 2: Results for experiments conducted using Imagenet-1k(mini) dataset

### 4.1.3 Do Counterfactual images improve the OOD robustness of the classifier?

**Classification Accuracy (MNIST Dataset)** Firstly, we trained a classifier on counterfactuals generated by the pre-trained CGN provided by the authors. It was not clear how many counterfactual images the classifier should be trained on, but the accuracies in Table 3 were similar to the results in the ablation study in Fig. 7 using $10^6$ counterfactuals, so this is the number we chose. There was also ambiguity between the statements in the paper and the code about the classifier being trained on any real images, so we trained two classifiers. One classifier was shown real images, and the other was not.

The classifier trained with counterfactuals generated by the pre-trained models achieved comparable results to those in

| | Colored MNIST | | Double-colored MNIST | | Wildlife MNIST | |
|---|---|---|---|---|---|---|
| | Train Acc | Test Acc | Train Acc | Test Acc | Train Acc | Test Acc |
| Pre-Trained (Ours/With real images) | 100.0 | 96.98 | 98.9 | 92.29 | 99.7 | 88.35 |
| Pre-Trained (Ours/Without real images) | 100.0 | 92.70 | 98.9 | 90.42 | 99.8 | 85.09 |
| Trained (Ours/With real images) | 98.7 | 68.96 | 96.8 | 88.54 | 99.9 | 72.93 |
| Trained (Ours/Without real images) | 98.7 | 43.88 | 96.7 | 87.90 | 99.9 | 75.28 |
| Original+CGN (Theirs) | 99.7 | 95.10 | 97.4 | 89.00 | 99.2 | 85.70 |

Table 3: MNIST Classification Accuracy

the paper. From table 3, it can be seen that the pre-trained models achieved train accuracies that differed by less than 3%, and test less than 1.5% compared to the results in the paper. However, the classifier trained on counterfactuals generated by CGNs that we trained (using the provided configurations) performed significantly worse on colored MNIST and

wildlife MNIST in terms of test accuracy. We anticipate that the provided configurations were not the same as the configurations used to acquire the results in the paper.

The presence of real images in the dataset for the pretrained models appeared not to have a significant effect on train or test accuracy. The largest gain obtained by including real images was approximately 4%. This demonstrates that the ambiguity regarding whether or not real images were used in the training of the classifier was inconsequential. For the CGNs that we trained, however, the presence of real images improved the performance of the classifier significantly.

**Classification Accuracy(ImageNet Dataset)**

The classifier was trained on counterfactual images from pre-trained CGN and ImageNet-1k(mini). The results in table 2(c) indicate the trend that was observed. The training accuracy showed a similar trend to the original paper's classifier (trained on ImageNet). There is a similar drop in the training accuracy compared to the baseline(ImageNet-1k).

Even though the original paper does not include the test accuracy for the classifier for the same distribution, we found that the classifier does not perform well with respect to the test data. The drop in top-1(the predicted class is the correct class that the image corresponds to) & top-5(5 out of 1000 classes with the highest probability as predicted by the classifier matches the actual label) accuracy compared to the baseline was attributed to the ability of the counterfactual models to reduce the shape bias of classifier which would improve the classifier's robustness to unseen data. However, this is invalidated by the low percentage of the test accuracy. To further understand why the classifier ensemble is not performing well with unseen test data, we conducted additional experiments to explain the same behaviour.

**Out of distribution accuracy**: A similar study as given in the paper was conducted to understand how the trained model performs with an out-of-distribution dataset. Table 2(b) contains the information with respect to the ImageNet-1k(mini) + CGN. There is a significant reduction in the accuracy of the out-of-distribution dataset. The baseline also showed a similar trend, and we could not achieve the higher percentage reported as part of the paper. We concluded that the baseline result is on the lower side primarily because of the size of the ImageNet-1k(mini) dataset that was used for training. Since the results show that the ensemble classifier improves the out-of-distribution robustness compared to the baseline, the percentage was still very low to make any conclusion.

Both the trend with the test accuracy and out-of-distribution accuracy falls on the lower side, which prompted us to investigate further. We generated explainability plots using the same distribution and out-of-distribution data to determine how the model is behaving with and without the heads that disentangle shape, texture, background. We recorded All of the experiments as part of section 4.2.

## 4.2   Results beyond original paper

**For Additional Result 1 we make use of CGN[11] architecture that has been designed for MNIST datasets due to computational limitations.**

### 4.2.1   Additional Result 1 - Does $f_{bg}$, $f_{texture}$, $f_{shape}$ and $\mathcal{L}_{perceptual}$ (Perceptual loss) proposed in [11] cover all aspects of background, shape, texture?

CGN [11] makes use of texture loss $\mathcal{L}_{text}$ $(x_{gt}, x_{gen})$, = sampling 36 patches of size 15 x 15 grid from regions wherever mask has values near 1. Further, from these 36 patches, a patch grid of 6 x 6 is used. It is then upscaled to 256 x 256 resolution, which is in turn used an input to the Perceptual loss $\mathcal{L}_{perceptual}$ between foreground f and patch grid $\mathcal{L}_{text}(f, pg)$. However, we observe that important image properties such as luminance, contrast, structure are not taken into consideration with the $\mathcal{L}_{text}$ loss proposed in CGN [11] for the generated image and the ground truth image and also because

Hence, we propose the usage of an additional Loss function $\mathcal{L}_{ssim}$ (SSIM) [12]. In addition, motivated by results as shown in [13], [7] L2 loss unlike SSIM [12] over different distortions of the image remains constant instead of recognising them . It complements the structural loss $\mathcal{L}_{rec}$. Default Gaussian Kernel of 11 was used as a hyperparameter for SSIM [12].

We observe from Table 4 that using SSIM [12] loss improves classification accuracy on the Wildlife MNIST dataset. Qualitative improvements in the generated images can be seen in Fig 3b. Images trained with SSIM [12] loss show better structure and crisper outlines. Improvements can be seen using SSIM [12] loss on the Double Colored MNIST dataset to a lesser extent. However, accuracy on the colored MNIST dataset decreases. This may be due to the dataset's

shape/structure/bias. Comparing Fig. 3a and Fig. 3b, we observe that usage SSIM [12] leads to generation of mask

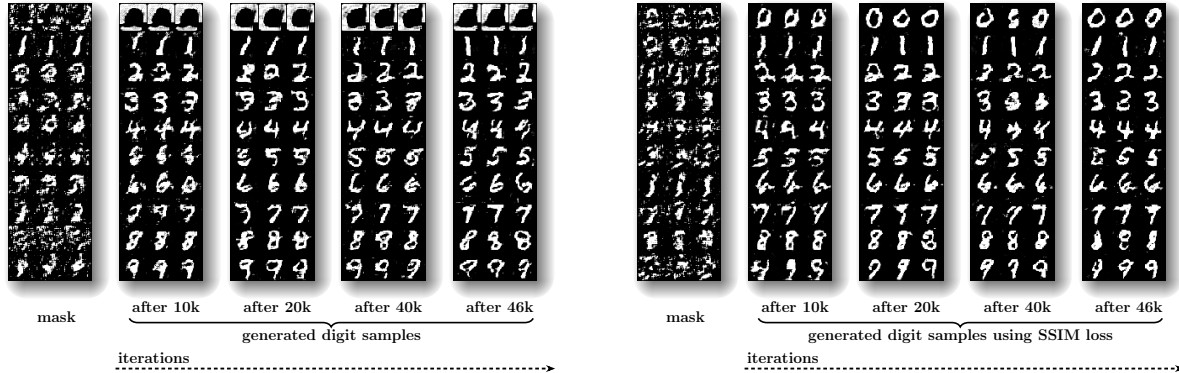

(a) Wildlife MNIST mask samples obtained using default hyper-parameters mentioned in CGN [11].

(b) Wildlife MNIST mask samples obtained by adding SSIM [12] loss.

Figure 3: Results for experiments conducted using Wildlife MNIST dataset

samples that are sharper, capture more structure details, clearer outputs. Specifically, when compared to Fig. 3a the digits 0, 2, 4 lead to better visual outputs. As a result, we show in Table 1 that the overall classifier's accuracy increases by around 16% when compared to training from scratch by us, and around 6% when compared to accuracy of given pre-trained model.

| Datasets | Using pretrained weights | Training from scratch | Trained from scratch with SSIM [12] |
|---|---|---|---|
| Colored MNIST | 96.42 | 61.12 | 44.77 |
| Double Colored MNIST | 86.26 | 86.19 | 87.88 |
| Wildlife MNIST | 71.89 | 61.94 | 77.64 |

Table 4: Accuracy for MNIST datasets when SSIM [12] loss function is used. For the Wildlife dataset and Double colored dataset we observe an increase in the overall accuracy when compared to what has been reported in the paper with the usage of SSIM [12]

### 4.2.2 Exploring classifier robustness with ImageNet

From 2(c), we find a considerable drop in the training and test accuracies(top-1) compared to the baseline. To explain the performance of the model, we integrated lime[9] package to generate explainability heatmap plots.(code reference *lime_plots.py*)

**Same distribution Test set** Fig 4 shows the outcome of the plots using the same image(from an unseen set) run through 2 different classifiers. Firstly, we used a pre-trained Resnet-50 to find out the robustness of the same towards unseen dataset. Secondly, we made use of a fully trained classifier ensemble with a pre-trained Resnet-50 as the backbone and 3 different heads as specified in the original paper[11]. The results are recorded by obtaining the top-5 classes with highest probability.

The image on the left of Fig 4 was classified as 'iPod' with regions including the object and the background contributing towards it. The plot shows how the classifier is extracting information from not only the object but also the background to determine the correct class. On the other hand, the image on the right shows the explainability plot when the suggested classifier ensemble is used. It performs poorly categorising the image as 'American_chameleon' with a higher probability when compared to the actual classification 'iPod'. The heatmap sheds the light into this behavior showing that the classifier does not include the background(as evident from the red zone) and focuses primarily on the object shape to make a decision.

From the above experiments through visual plots, we are able to determine that the counterfactual images to skew the shape-bias of the classifier does not contribute to the robustness towards unseen data within the same distribution. This

can be attributed to the inclusion of counterfactual images that are of reduced realism which affects the classifier from learning meaningful information from the dataset at hand.

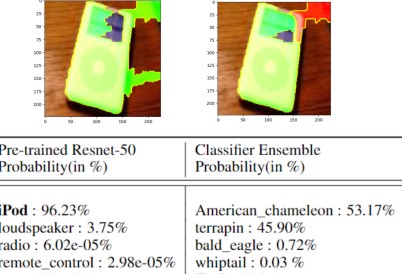

Figure 4: Heatmap plots and corresponding classification(probability in %) of the top 5 best classes for the image iPod. From left to right, same image classified with a pre-trained Resnet-50 & Classifier ensemble architecture from the original paper[11]. Green regions contribute towards the classification while red regions do not.

## 5 Discussion

### 5.1 What was easy

It was easy to set up the environment as listed/indicated in the README file of the Github repository. Although not all commands were explicitly listed, it helped us navigate through and run the code. The presence of .yaml files for each dataset in the case of MNIST [6] helped us to train CGNs and classifiers with well-working hyperparameters quickly.

ImageNet experiments were structured clearly in multiple sections within the codebase. It made it easier to understand the difference in the architecture that was followed to handle Mnist, ImageNet. Since, reliance on pre-trained network for ImageNet was important, the presence of scripts to download all the data, weights made the setup easier.

### 5.2 What was difficult

In the case of the architecture for ImageNet, replacing it with ImageNet-1k or Mini-ImageNet required code changes. The python parameters to load the dataset(–data) had no effect that prompted changes in the dataloader.py. The classifier(*train_classifier.py*) did not have provision to generate the values without mandatorily providing the counterfactual information. This proved to be a challenge as we needed the baseline results to compare the performance of the proposed model. Code modification was done to accommodate the same and the experiment was conducted.

The results from the original paper included the inception score for the proposed CGN, but we could not find a code block to calculate the same. Considerable amount was spent on trying to find out the hyperparameters that was needed to generate the counterfactual images. Since the inception score was dependent on the number of counterfactuals generated, we worked towards identifying the correct hyperparameters before continuing with classifier training.

**Can the generative model be trained on a single GPU?** From table 5, we were able to train the generative model from scratch for all variations of MNIST. However, for Imagenet architecture, with the default parameters, it was going to take upwards of 200 hours. Therefore, we were unable to verify this claim.

### 5.3 Suggestions for reproducibility

In general, the resources provided by the authors on GitHub in conjunction with the explanations in the paper were sufficient to generate similar results to those found in the paper with relative ease. However, in the future, it may be helpful if the authors provided the weights of the exact models used in the paper, along with the hyperparameters used to train them.

In addition, the size of the ImageNet dataset makes running several experiments infeasible without significant compute power. Therefore, we suggest that additional experiments using a subset of ImageNet (i.e. Mini-ImageNet) be added to the report for the sake of reproducibility.

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

# Appendices

## A Ablation Study

We conducted experiments to recreate the MNIST Ablation Study. For this study, the pre-trained model provided by the authors was used. We observed a similar trend to the authors. An increase in the number of counterfactual images used in training resulted in higher training accuracies. However, our values differed significantly from those in the report, as seen in Fig. 5 and Fig. 6 . In particular, we observed higher accuracies for each dataset, especially when only $10^4$ counterfactuals were used in training. This difference may be explained by differences between the pre-trained models provided by the authors and the models that were used to generate the plots.

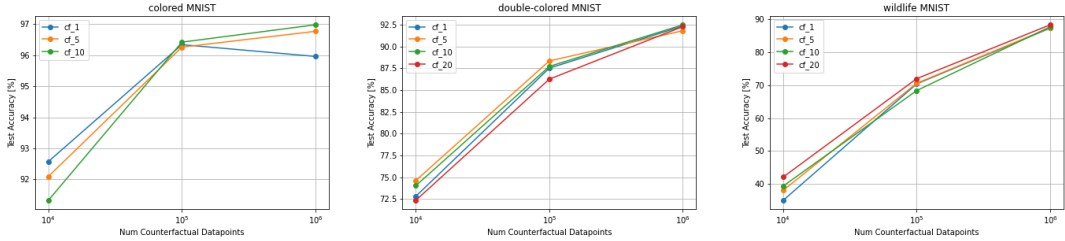

Figure 5: Recreated MNIST Ablation Study

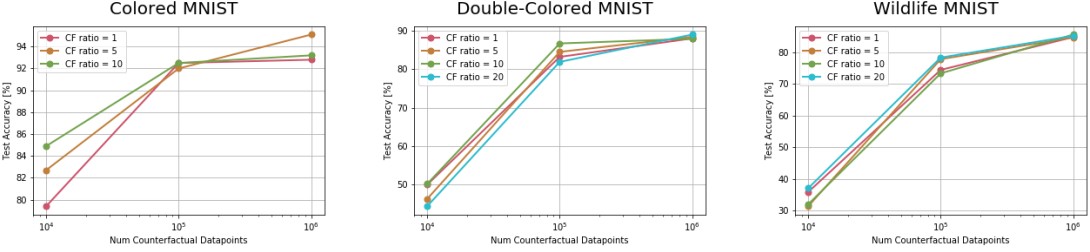

Figure 6: Original MNIST Ablation Study from CGN[11]

## B Training time for Generative Model

The following table shows the training time for each generative network against the dataset that was used.

Note: The Imagenet based CGN depends only on the biggan-256 backbone and U2-net to train. The MNIST based CGN architecture however, trains using the dataset without any pre-trained weight as backbone. Imagenet counterfactual generation was going to run for 1.2 million iterations(0.5s/iteration), which was not computationally feasible with our resources.

| Dataset | Training time (in hours) |
|---|---|
| Colored MNIST | $\approx 0.6$ |
| Double-colored MNIST | $\approx 0.6$ |
| Wildlife MNIST | $\approx 3.5$ |
| Imagenet | $\approx 167$ |

Table 5: Training time for CGN for different datasets

# C   Counterfactual Images

278 The following images using the pre-trained CGN model that was provided with the codebase. Minor deviations were observed with the image given in the paper to the result we obtained.

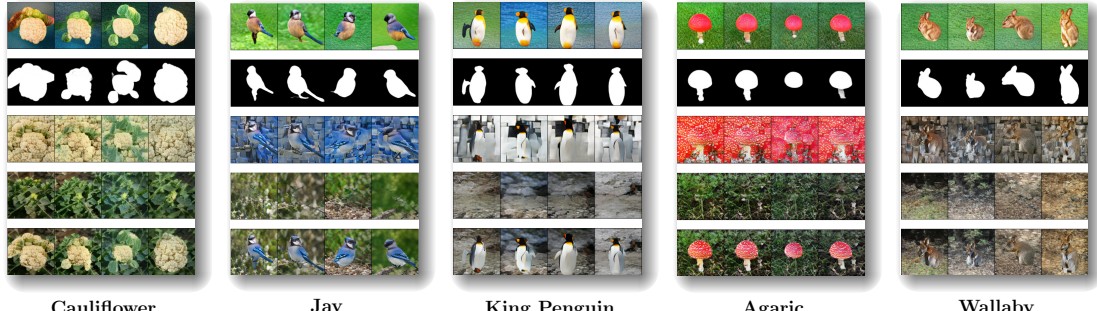

| Cauliflower | Jay | King Penguin | Agaric | Wallaby |

Figure 7: Grid of Counterfactual Images from the Pre-trained CGN [11] as given in the original paper. The CGN is trained with biggan-256 as the backbone and Pre-trained U2-net for mask generation.

279

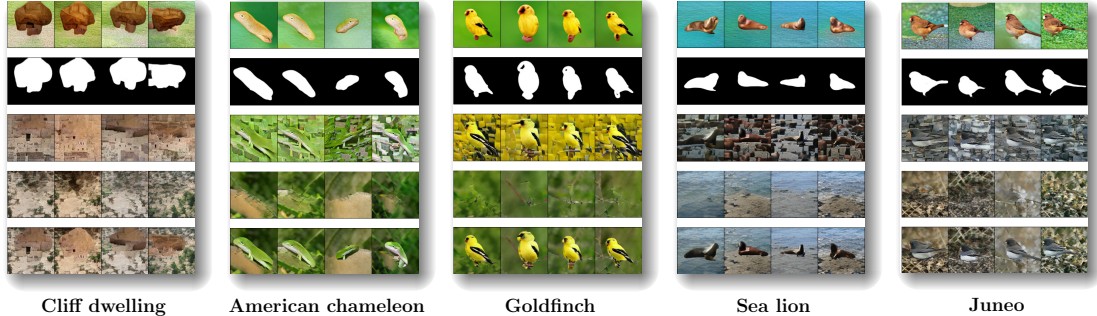

| Cliff dwelling | American chameleon | Goldfinch | Sea lion | Juneo |

Figure 8: Grid of Counterfactual Images from same class that have poorer xgen. All classes are picked at random and the counterfactual analysed for 'realism'

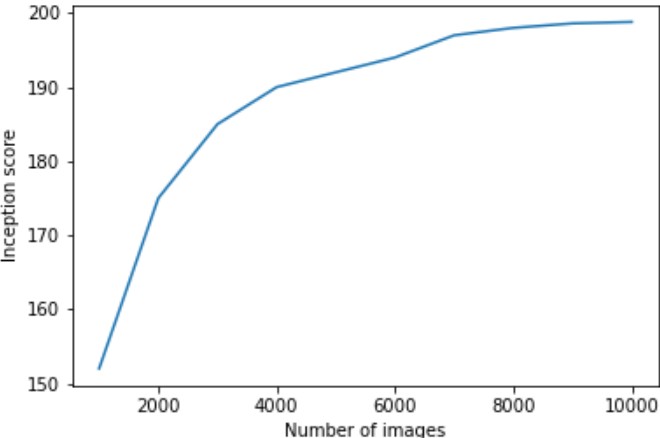

Figure 9: Inception score (10 splits) of images generated by the pre-trained CGN

## D SSIM Loss function

SSIM [12] helps preserve the structural properties between the two images by using luminance, contrast and structural information. Additionally, SSIM [12] leads to generating better structured masks using the 'm' that helps to localize the digits in a better way in the final output $x_{gen}$.

SSIM [12] is defined using the three aspects of similarities, luminance $\big(l(x, x_{gen})\big)$, contrast $\big(c(x, x_{gen})\big)$ and structure $\big(s(x, x_{gen})\big)$ that are measured for a pair of images $\{x, x_{gen}\}$ as follows: Given two images ground truth $x$ and generated image $x_{gen}$, the SSIM [12] loss is defined [7] as follows:

$$\mathcal{L}_{ssim}(\alpha) = 1 - \mathbf{E}_x\left[l(\alpha).cs(\alpha)\right] \tag{1}$$

$$l(x, x_{gen}) = \frac{2\mu_x\mu_{x_{gen}} + C_1}{\mu_x^2 + \mu_{x_{gen}}^2 + C_1} \tag{2}$$

$$c(x, x_{gen}) = \frac{2\sigma_x\sigma_{x_{gen}} + C_2}{\sigma_x{}^2 + \sigma_{x_{gen}}{}^2 + C_2} \tag{3}$$

$$s(x, x_{gen}) = \frac{\sigma_{xx_{gen}} + C_3}{\sigma_x\sigma_{x_{gen}} + C_3} \tag{4}$$

where $\mu$'s denote sample means and $\sigma$'s denote variances. $C_1, C_2$ and $C_3$ are constants. With these, SSIM and the corresponding loss function $\mathcal{L}_{ssim}$, for a pair of images $\{x, x_{gen}\}$ are defined as:

$$SSIM(x, x_{gen}) = l(x, x_{gen})^\alpha \cdot c(x, x_{gen})^\beta \cdot s(x, x_{gen})^\gamma \tag{5}$$

where $\alpha > 0$, $\beta > 0$ and $\gamma > 0$ are parameters used to adjust the relative importance of the three components.

$$\mathcal{L}_{ssim}(x, x_{gen}) = 1 - SSIM(x, x_{gen}) \tag{6}$$

### D.0.1 Additional Result 2 - Exploring the biased behaviour of CGN[11] with the datasets

To investigate the robustness of the CGN architecture [11] to varied color augmentations, we applied color jitter to augment the training data. We found that applying a color jitter decreased classification accuracy by 10% on double-colored MNIST and 50% on wildlife MNIST.

Amongst all widely known augmentations we make use of color jitter since from [2], [5] it is evident that color jitter, sobel flter augmentations are imperative to learn useful representations from the given dataset.

We observe that from Table 6 that when we used it on Double Colored dataset the classifier's accuracy decreases by almost 10 %. Similarly, there is decrease in accuracy of Wildlife MNIST dataset by almost around 50% as indicated in Table 4.

| Datasets | Using pretrained weights | Training from scratch | Trained from scratch using jitter |
|---|---|---|---|
| Double Colored | 86.26 | 86.19 | 78.56 |
| Wildlife | 71.89 | 61.94 | 10 |

Table 6: Accuracy for MNIST datasets when Color Jitter augmentation is used.

To determine why the color jitter augmentation decreases training accuracy, we observed the results visually through the samples generated across 40K iterations by the CGN. It can be seen that digit 6 loses its shape over iterations. Digits 0 and 1 have the same background and similar digit font. These artefacts produced by the CGN[11] are a likely cause of the classifier's decreased performance. Which might indicate that the CGN is overfitting itself to the image backgrounds while learning the generative model cGAN using the loss functions.

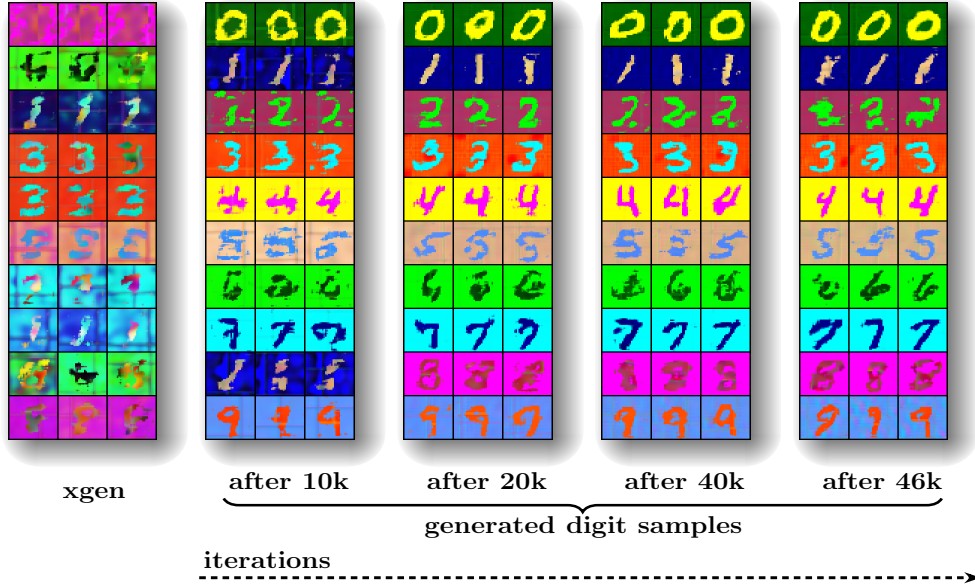

xgen      after 10k    after 20k    after 40k    after 46k

generated digit samples

iterations

Figure 10: Double Colored MNIST samples obtained using default hyper-parameters mentioned in CGN [11].

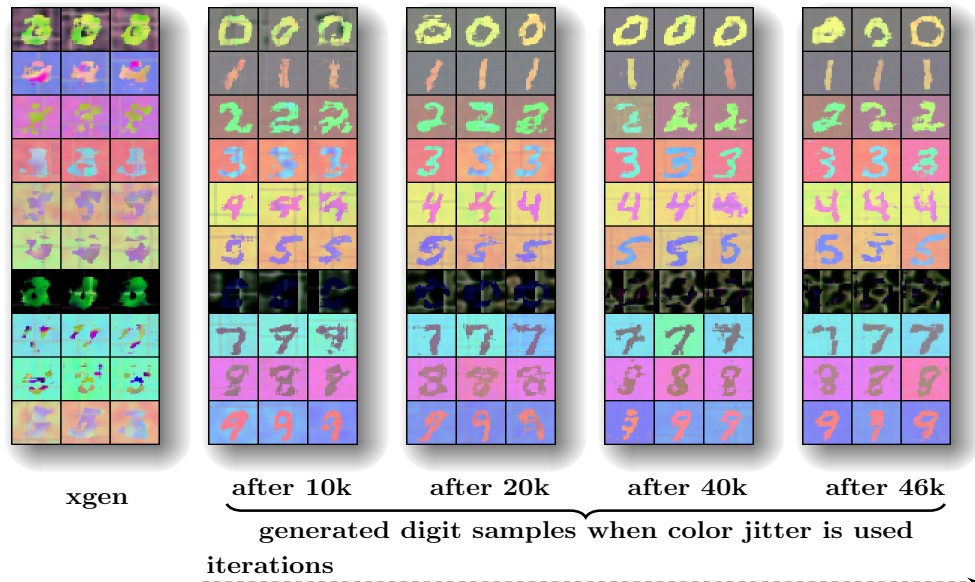

xgen      after 10k    after 20k    after 40k    after 46k

generated digit samples when color jitter is used

iterations

Figure 11: Double Colored MNIST samples obtained using addition of color jitter. We observe that it leads to generation of samples that are not indicative of the actual samples from the Double Colored MNIST dataset. We observe that there is difference between with/without augmentation in terms of the brightness, contrast, overall image representations. Specifically, digit 6 loses its shape, texture, colors. Similarly, digits 0,1 are generated using different colors in contrast to Fig. 10. Therefore, the visual samples indicate possibly why the classifier's accuracy drops by around 10%.

