# OpenReview forum: "Counterfactual Generative Networks"
_ML_Reproducibility_Challenge/2021/Fall — RC2021_

### Official Review · Reviewer_YqBk · 2022-02-28
**A detailed report with extensive experimental results and additional improvments**

**Rating:** 9
**Confidence:** 4

**Review:**

This report provided a comprehensive analysis of the counterfactual generative network. The reproduced results are shown to be a good complementary and in-depth analysis of the original paper of the counterfactual generative network. Here I provide some details on the pros and cons of this report. Please feel free to correct me if I made anything wrong.

### **Pros**:
1. The analysis is comprehensive, inspiring, and in-depth. Specifically, the analysis in Sec 4.2 really helps the community to gain a better understanding of the original work. Meanwhile, the experiments are extensive, and the listed hyper-params can help other researchers to reproduce the work as well.

2. The proposed method beyond the original paper is also very inspiring and the empirical results indicate the improvements by the additional SSIM loss.

### **Cons**:
1. More detailed or formalized analysis of SSIM loss can be added in Sec 4.2. I have an idea about the SSIM, which would help the algorithm to learn the structural properties. However, it would be even better if a more elaborate analysis can be provided in the main paper. I noticed that there has been some analysis in Sec D in the supplementary file thus the authors can consider moving some crucial parts into the main paper for better readability.

2. It is better to provide also the std to show the stability of each method in Table 4.

3. Some suggestions to further improve the readability: **(a)** ImageNet-1k or ImageNet 1000? It seems that two presentations occurred in the report. Try to be more unified; **(b)** In Sec 3, it would be better to provide some description on different losses, including $L_\text{rec}$, and $L_\text{perceptual}$; **(c)** It would be better if the authors can make a few equations or symbols to be wrapped with "*" among the text; **(d)** Some tables (e.g., Table 1-2) are missing the bottom lines.

Overall, the report is a strong submission and I would suggest it can be accepted into this workshop. The authors could probably further improve it based on the suggestions I proposed.

---

### Official Review · Reviewer_i3DX · 2022-03-05
**reproduce results using the paper's code with modifications**

**Rating:** 6
**Confidence:** 3

**Review:**

This paper carefully reproduces the results of paper using the original implementation to validate several claims in the paper. It also perform exploration analysis on the claims. In addition, this paper propose to use an extra loss function to mitigate some of the shortcomings during counterfactual generation process and generate heatmap plots to study classify behavior. The reproduced results are good.

---

### Official Review · Reviewer_uyV3 · 2022-03-19

**Rating:** 7
**Confidence:** 4

**Review:**

The paper describes the reproducibility of the "counterfactual generative networks" paper. Here are my comments.

1. The paper is clearly written and easy to follow. The authors have gave a nice summary of the counterfactual generative networks paper in the introduction, followed by different sections that describes the scope, methodology, experiments, etc.

2. The authors have conducted thorough experiments based on the original code provided by the original authors. Authors made neccessary changes to the code for experimentation on small imagenet datasets (due to compute limitations). The experimental results are discussed thoroughly. Authors also performed additional experiments in order to analysis the results.

Overall, the authors did a good job reproducing the results and went above and beyond to understand the model. The paper is also well written and easy to follow.

---

### Meta-Review · Area_Chair_5nGR · 2022-04-08

**Recommendation:** Accept
**Confidence:** 4

**Metareview:**

Overall a good reproducibility study. A more detailed analysis and improvements suggested by the reviewers would be a good addition to the paper.

---

### Decision · Program_Chairs · 2022-04-09

**Decision:**

Accept

**Comment:**

Following the recommendation of reviewers and meta-reviewer, the paper is accepted for ML Reproducibility Challenge 2021, and will be published in the upcoming special edition of ReScience Journal.